# The Impact of COVID-19 on the Epidemiology and Outcomes of Candidemia: A Retrospective Study from a Tertiary Care Center in Lebanon

**DOI:** 10.3390/jof9070769

**Published:** 2023-07-21

**Authors:** Aline El Zakhem, Omar Mahmoud, Johnny Zakhour, Sarah B. Nahhal, Nour El Ghawi, Nadine Omran, Walaa G. El Sheikh, Hani Tamim, Souha S. Kanj

**Affiliations:** 1Division of Infectious Diseases, American University of Beirut Medical Center, Beirut 110236, Lebanon; az51@aub.edu.lb (A.E.Z.); njo01@mail.aub.edu (N.O.); 2Biostatistics Unit, Clinical Research Institute, American University of Beirut, Beirut 110236, Lebanonhtamim@aub.edu.lb (H.T.); 3College of Medicine, Alfaisal University, Riyadh 11533, Saudi Arabia; 4Center for Infectious Diseases Research, American University of Beirut, Beirut 110236, Lebanon

**Keywords:** *Candida*, candidemia, invasive candidiasis, *Candida auris*, non-albicans *Candida*, COVID-19, antifungal susceptibility, Lebanon, Arab world, EQUAL score

## Abstract

Invasive fungal infections, notably candidemia, have been associated with COVID-19. The epidemiology of candidemia has significantly changed during the COVID-19 pandemic. We aim to identify the microbiological profile, resistance rates, and outcomes of COVID-19-associated candidemia (CAC) compared to patients with candidemia not associated with COVID-19. We retrospectively collected data on patients with candidemia admitted to the American University of Beirut Medical Center between 2004 and 2022. We compared the epidemiology of candidemia during and prior to the COVID-19 pandemic. Additionally, we compared the outcomes of critically ill patients with CAC to those with candidemia without COVID-19 from March 2020 till March 2022. Among 245 candidemia episodes, 156 occurred prior to the pandemic and 89 during the pandemic. Of the latter, 39 (43.8%) were CAC, most of which (82%) were reported from intensive care units (ICU). Non-albicans *Candida* (NAC) spp. were predominant throughout the study period (67.7%). *Candida auris* infection was the most common cause of NAC spp. in CAC. *C. glabrata* had decreased susceptibility rates to fluconazole and caspofungin during the pandemic period (46.1% and 38.4%, respectively). The mortality rate in the overall ICU population during the pandemic was 76.6%, much higher than the previously reported candidemia mortality rate observed in studies involving ICU patients. There was no significant difference in 30-day mortality between CAC and non-CAC (75.0% vs. 78.1%; *p* = 0.76). Performing ophthalmic examination (*p* = 0.002), CVC removal during the 48 h following the candidemia (*p* = 0.008) and speciation (*p* = 0.028) were significantly associated with a lower case-fatality rate. The epidemiology of candidemia has been significantly affected by the COVID-19 pandemic at our center. Rigorous infection control measures and proper antifungal stewardship are essential to combat highly resistant species such as *C. auris*.

## 1. Introduction

*Candida* spp. are the fourth most commonly isolated pathogens in nosocomial bloodstream infections (BSI) in the United States and are associated with over 350,000 yearly deaths worldwide [1]. The epidemiology of candidemia varies widely around the globe. Studies from Lebanon show a predominance of non-albicans *Candida* (NAC) spp. in episodes of candidemia [2], which is similar to what has been reported from European countries [3] and other neighboring countries from the Middle East region [4,5,6].

Since December 2019, the COVID-19 pandemic has triggered a global health crisis. Critically ill COVID-19 patients have been reported to be more susceptible to co-infections with other pathogens, notably multidrug-resistant (MDR) bacterial and fungal organisms [7,8]. The incidence of COVID-19-associated candidemia (CAC) has been highly variable from different reports, ranging from 0.03% to 9% [9]. Both the incidence and mortality of candidemia in patients with COVID-19 is substantially higher compared to patients without COVID-19 and may reach 75% [10]. Additionally, the onset of candidemia is earlier in patients with COVID-19 compared to patients without [11,12]. CAC also results in prolonged length of hospitalization and overall worse patients’ outcomes [12].

Although one study mentioned that there were no significant differences in the microbiological profile of candidemia in patients with COVID-19 and those without, other reports confirm that NAC spp. appear to be the most predominant in CAC [10,13,14]. The emergence of highly resistant *Candida auris* during the COVID-19 pandemic has been of particular concern and has been responsible for multiple outbreaks in healthcare settings, especially in intensive care units (ICU), including in health care settings that had not reported such infections previously [8,10].

The aim of this study was to describe the changing epidemiology and rates of antifungal resistance of *Candida* spp. causing candidemia in a tertiary care center in Lebanon before and after the COVID-19 pandemic. Additionally, we aimed to compare the outcomes of patients with CAC and those with candidemia but not infected with COVID-19. We also investigated the impact of compliance with the elements of the European Confederation of Medical Mycology Quality of Clinical Candidemia Management (EQUAL) Candida score on mortality in patients with CAC. 

## 2. Research Design and Methods

### 2.1. Study Design and Setting

We conducted a retrospective study that included all patients admitted with candidemia to the American University of Beirut Medical Center (AUBMC) over a span of 18 years from January 2004 to March 2022, including the period of the COVID-19 pandemic in our region (1 March 2020, till 30 March 2022). One objective was to look at the mortality of patients with candidemia admitted to the ICU during the COVID-19 pandemic. AUBMC is a 420-bed academic tertiary-care center and a national and regional referral center in Beirut, Lebanon. AUBMC receives over 25,000 inpatient admissions annually. It provides specialized medical and surgical services including oncology and bone marrow transplantation services. The Clinical Microbiology Laboratory (CML) at AUBMC is accredited by the College of American Pathologists since 2004 and uses the Clinical and Laboratory Standards Institute breakpoints (CLSI) breakpoints for antimicrobial susceptibility breakpoints.

### 2.2. Population and Data Collection

Using the Electronic Health Records (EHR) at AUBMC, we identified all confirmed episodes of candidemia from hospitalized patients between January 2004 and March 2022. Inclusion criteria were patients (1) aged 18 years old and above (2) who had candidemia defined by isolation of *Candida* spp. in at least one blood culture. We excluded episodes of candidemia with concomitant bacteremia. Patients whose outcomes were not available at 30 days were excluded from the mortality analysis. 

After determining the prevalence of each *Candida* spp. during each year and trends of antifungal susceptibility within each year, we sought to compare the outcomes of critically ill patients with CAC and those with candidemia without COVID-19. For this analysis, we included episodes of candidemia that occurred from March 2020 till March 2022. To account for the dynamic changes in *Candida* spp. epidemiology at our center during the pandemic, we focused on comparing outcomes in the subpopulation of patients admitted to the ICU only during the pandemic. The first group consisted of all episodes of candidemia that were not associated with COVID-19 (non-CAC), while the second group consisted of episodes of CAC. 

Data collected included demographics (age, sex), comorbidities (diabetes mellitus, hemodialysis), immunosuppression (malignancy, chemotherapy, or immunotherapy within the previous 30 days, hematopoietic or solid organ transplantation, neutropenia), and patient characteristics at the onset of infection (hospital unit, mechanical ventilation (MV), presence of central venous catheter (CVC), parenteral nutrition, abdominal surgery within the previous 30 days, antibiotic and antifungal history within the previous 30 days, and source of candidemia). We also extracted data regarding COVID-19 clinical course and management (including acute respiratory distress (ARDS), need for MV, treatment with corticosteroids, and immunomodulatory medications) for patients who were admitted between 1 March 2020, and 30 March 2022. We defined neutropenia as an absolute neutrophil count (ANC) <1500 cells/mm^3^, which was further categorized into mild (<1500 and ≥1000 cells/mm^3^), moderate (<1000 and ≥500 cells/mm^3^), and severe (<500 cells/mm^3^) [15].

### 2.3. Microbiological Definitions

All cases of COVID-19 were confirmed using a real-time polymerase chain reaction (RT-PCR) that was performed on nasal swabs, tracheal aspirates, broncho-alveolar lavage, or other respiratory samples. 

Candidemia was defined as the isolation of *Candida* spp. from blood culture of a peripheral or central sample. Non-albicans candidemia was defined as the isolation of a NAC spp. from initial blood cultures, whether it occurred individually or as co-infection with *C. albicans*. Blood culture bottles were incubated in a BACT/ALERT^®^ system (Durham, NC, USA), and *Candida* spp. identification and antifungal susceptibility testing were performed using VITEK^®^ 2 system version 8.01 (BioMérieux, Marcy L’Etoile, France). Additionally, *C. auris* colonies were phenotypically identified by Matrix-Assisted Laser Desorption/Ionization-Time of Flight MALDI-TOF (Bruker Daltonik, GmbH, Bremen, Germany). Antifungal susceptibility testing was conducted using VITEK-2 and E-test antimicrobial susceptibility tests. The microbiology laboratory used the CLSI M60 minimal inhibitory concentration (MIC) breakpoints for susceptibility testing of *Candida* spp. [16]. The interpretation of the minimum inhibitory concentrations (MICs) susceptibility breakpoints (μg/mL) for *C. auris* were based on Centers for Disease Control and Prevention (CDC) and Clinical and Laboratory Standards Institute (CLSI) guidelines, essentially defined based on those established for closely related *Candida* spp. (*Candida haemuloni*) and on expert opinion. In this context, the designated resistant breakpoints for *C. auris* are as follows: fluconazole, ≥32 μg/mL; anidulafungin, ≥4 μg/mL; caspofungin, ≥2 μg/mL; micafungin, ≥4 μg/mL; amphotericin B, ≥2 μg/mL. Voriconazole susceptibility breakpoints are not applicable, and it is recommended to consider using fluconazole susceptibility as a surrogate susceptibility assessment [17]. Resistant breakpoints to voriconazole were extrapolated from fluconazole as a surrogate susceptibility assessment. Due to the financial issues in view of the economic crisis in Lebanon and the COVID-19 pandemic, MICs were not requested on several isolates.

Candidemia was considered CAC if the episode occurred within 42 days following the onset of COVID-19. Recurrent candidemia was defined as two episodes of candidemia occurring ≥14 days apart with clinical and microbiological resolution in the interim.

We used the National Healthcare Safety Network (NHSN) definition of central line associated bloodstream infection (CLABSI) [18]. As per the Infectious Diseases Society of America (IDSA) guidelines for the treatment of invasive candidiasis, appropriate duration of treatment was defined as 14 days of antifungal treatment after the first negative blood culture [19]. 

### 2.4. Statistical Analysis

Data management and analysis were conducted using IBM SPSS version 28 (IBM, New York, NY, USA). Categorical data were presented using count (percent), while continuous data were presented using mean ± standard deviation (SD). Associations between categorical variables and the outcome variable (CAC vs. non-CAC) were assessed using the Chi-square test or Fisher’s exact test when ≥20% of expected cell counts are below 5. Associations between continuous variables and the outcome variable (CAC vs. non-CAC) were assessed using independent-samples t-test. Significance was set at p < 0.05.

### 2.5. Ethical Considerations

The study received approval by the institutional review board (IRB) at AUBMC (Protocol number: BIO-2019-0290). Patient consent was waived as this is a retrospective chart review. The study included all adult patients with candidemia presenting to AUBMC in the study period, with no regard to sex and ethnic background. It posed no risk to patients. The potential benefits of the study outweigh the potential risks.

## 3. Results

### 3.1. Baseline Characteristics of the Study Population

We identified 233 patients with a total of 245 episodes of candidemia between 2004 and 2022. We reported 245 episodes of candidemia of which 156 occurred prior to March 2020 and 89 during the pandemic. Among the latter, 39 (43.8%) were CAC while the remainder (56.2%) were reported from patients without COVID-19. Most episodes of CAC (82%) occurred in the ICU. In patients with CAC, the median time between SARS-CoV-2 positive testing and candidemia was 24 days while median time from ICU admission to candidemia was 17.5 days.

### 3.2. Epidemiology of Candida spp.

Over the span of 18 years, we found a predominance of NAC spp. (67.7%) compared to *C. albicans* (32.2%). NAC were predominant in both patients with and without COVID-19 (74.3% and 66.5%, respectively). We noted an increase in the prevalence of NAC spp. between 2020 and 2022. Speciation was conducted for 83.3% of all isolates. For NAC spp., the rate of speciation was 66.7% between 2004 and 2008, 70% between 2009 and 2013, 79.6% between 2014 and 2019, and 76.1% between the years 2020 and 2022. 

*Candida glabrata* was the most predominant NAC spp. in the pre-pandemic era (42.7%). However, during the pandemic period we observed a decrease in the prevalence of *C. glabrata* (21.7%). We also reported the emergence of *C. auris* candidemia for the first time in Lebanon. After the pandemic, *C. auris* was responsible for 26 episodes of candidemia. Its prevalence was 32% in patients with non-CAC and 25.6% in patients with CAC (Figure 1). 

### 3.3. Antifungal Susceptibility

Antifungal susceptibility testing was performed on 140 isolates (57.1%) including 48.7% of *C. albicans* and 70% of NAC.

Overall, *C. albicans* isolates had high susceptibility rates to fluconazole (91.7%), voriconazole (88.2%), amphotericin B (100%), and caspofungin (100%). On the other hand, *C. glabrata* isolates had lower rates of susceptibility to fluconazole (58%), voriconazole (84%), and caspofungin (63.6%) but were highly susceptible to amphotericin B (97.2%) (Table 1).

Susceptibility results were available for twelve isolates of *C. auris* all of which were found to be resistant to azoles and amphotericin B but susceptible to caspofungin. 

### 3.4. Outcomes of CAC and Non-CAC

We compared characteristics, comorbidities, management, and outcomes of CAC and non-CAC in critically ill patients between March 2020 and March 2022. All patients were admitted to the ICU. None of the patients were found to have recurrent candidemia. We found no significant differences in age, sex, and comorbidities between both groups (Table 2). There were also no significant differences in the percentage of episodes occurring while patients were mechanically ventilated, had CVC, or had recently received antimicrobials including antifungals. We also found no significant differences in the epidemiology of *Candida* spp. between both groups. 

We noted a 30-day mortality of 76.6% from all episodes, with no significant difference between CAC and non-CAC. Regarding the management of both groups, we noted a significantly higher rate of empirical antifungal treatment with caspofungin for CAC compared to non-CAC (83.3% vs. 37.9%; P < 0.001). However, there was no statistically significant difference in empirical treatment with anidulafungin (P = 0.19) or micafungin (P = 0.49) between the two groups. Overall, the use of echinocandin as treatment was significantly higher in CAC compared to non-CAC (93.3% vs. 69.0%; P = 0.016) (Table 3). 

We found that 17% of all episodes of candidemia were treated for a duration of 14 or more days after the first negative blood culture. The median duration of treatment for CAC patients was 9.63 days and 16.06 days for non-CAC (P = 0.14). Moreover, our analysis showed that 79% of patients with CAC died before the completion of 14 days of antifungal treatment.

### 3.5. EQUAL Score Analysis

The mean EQUAL score in the total ICU population during the pandemic was not statistically significant between survivors and non-survivors (9.92 ± 3.82 vs. 9.12 ± 3.49, P = 0.28). Similarly, no statistical difference was found in scores in the subset of patients with CVC between survivors and non-survivors (10.36 ± 3.66 vs. 9.43 ± 3.36, P = 0.25). 

When assessing the components of the EQUAL score individually, survival was higher with susceptibility testing (35.7%) compared to episodes when susceptibility testing was not performed (16.7%). However, this was not statistically significant (P = 0.098). Although performing echocardiography was not associated with lower mortality, survival was significantly higher when ophthalmic examination was conducted (63.6% vs. 15.4%; P = 0.002). Survival was also higher when a CVC, if placed at onset of infection, was removed within 48 h (37.9% vs. 7.7%; P = 0.008), and when *Candida* speciation was performed (100% vs. 73.5%; P = 0.028).

Only 17% of our patient population completed the 14 days of antifungal treatment after the first negative blood culture with a significantly higher survival rate than those treated for a shorter duration (40% vs. 7.9%; P = 0.011).

## 4. Discussion

This study describes the changing epidemiology of candidemia over 18 years from a tertiary care center in Lebanon. It is also the first investigation of CAC in Lebanon. Only one study has been previously reported from the Arab countries in the Middle East region [10]. We observed notable differences in the epidemiology of candidemia during the COVID-19 pandemic compared to the pre-pandemic era. Our findings revealed the emergence of new *Candida* spp. with higher resistance rates during the pandemic. While we did not find evidence of higher mortality among patients with CAC compared to those with non-CAC, the management strategies differed between the two groups. We also noted a higher mortality rate of patients with candidemia during the pandemic compared to studies from the pre-pandemic period.

According to our analysis, we observed a significant change in candidemia epidemiology. In the pre-pandemic era, the microbiologic profile of candidemia at AUBMC was consistent with worldwide and Arab world observations regarding the increasing predominance of NAC [5]. Additionally, during the pre-pandemic period, *C. glabrata* was identified as the primary NAC spp.at our center [2]. However, during the pandemic period, we observed a decrease in *C. glabrata* prevalence with the emergence of *C. auris*, a highly resistant spp. that can rapidly colonize patients and spread in the units. This observation may differ from other studies conducted in Iran and Oman, where *C. albicans* and *C. glabrata* remained the predominant pathogens causing candidemia during the pandemic [9,20]. The COVID-19 pandemic overwhelmed the healthcare system, leading to breaches in infection control practices and measures [21]. This resulted in the transmission of *C. auris* and other MDR organisms among hospitalized patients [8,22], leading to multiple outbreaks worldwide, particularly in ICUs [23]. At our center, a significant number of *C. auris* candidemia cases were observed in 2021, indicating the occurrence of an outbreak resulting from intra-hospital transmission. In a previously published study from AUBMC by Reslan et al., genome sequencing analysis to determine clade distribution and antifungal resistance genes was performed on 29 isolates from different culture sites (blood, central venous catheter, deep tracheal aspiration, urine…). Whole-genome sequencing (WGS) using long reads sequencing (PacBio) was employed [24]. This study confirmed that all *C. auris* genomes belonged to the South Asian clade I among the five known global clades, regardless of the recovered source, site of specimen, or time span between isolations, thereby highly reflecting an outbreak due to hospital-associated transmission. However, due to financial constraints caused by the economic crisis in Lebanon and the ongoing COVID-19 pandemic, conducting genome sequencing analysis on all isolates was not possible.

We did not observe any difference in *Candida* spp. between CAC and non-CAC, as both groups had a higher prevalence of NAC. These findings are consistent with a previous study by Machado et al., which suggests that changes in the prevalence of MDR *Candida* spp. are not exclusive to COVID-19 patients, but may be linked to the increased use of antifungal medications [25]. In fact, COVID-19 affected the pattern of antifungal use in our center, with a significant increase in the use of echinocandins as first line antifungal treatment in CAC patients, probably due to the emergence of fluconazole resistant spp. such as *C. glabrata* and *C. auris* [26]. Such species are known to be more prevalent in COVID-19 ICU patients, particularly in those with prolonged length of stay [14,27]. Additionally, as per the international guidelines, echinocandins were used as first line therapy in the management of candidemia in the ICU including in patients with septic shock, and those with liver injury [27]. The unrestricted use of antifungals during the pandemic could have contributed to the increase in resistance rates of *Candida* spp. In our center [28]. Susceptibility rates to fluconazole in *C. glabrata* isolates decreased from 62% before the pandemic to 46.1% during the pandemic. In addition, there are increasing reports of *C. glabrata* resistance to echinocandins [29,30]. Our results show a rate of resistance of *C. glabrata* to caspofungin of 36.4%. Similar findings have been reported in studies from Turkey and Kuwait, which have shown an increase in the prevalence of MDR *Candida* spp. Among CAC patients [11,29,30]. The increased resistance rates are of high clinical significance, as echinocandins are considered first-line agents for empiric treatment of invasive candidiasis [19]. Our results show that 34.4% of patients with candidemia had received antifungal agents in the 30 days prior to onset of candidemia. These findings reiterate the importance of antifungal stewardship efforts to control the evolution of antifungal resistance among *Candida* spp. [31]. Furthermore it reinforces the need for the One-Health approach that expands antifungal stewardship efforts beyond human medicine only [32,33].

In our study, we used the EQUAL score to assess compliance with the guidelines of invasive candidiasis during the pandemic and to identify factors that may be associated with an increased 30-day mortality. Our results show that survivors had slightly higher EQUAL scores than non-survivors (9.92 ± 3.82 vs. 9.12 ± 3.49, P = 0.28) in the overall population as well as in CVC carriers (10.36 ± 3.66 vs. 9.43 ± 3.36, P = 0.25); however, the difference in scores was not statistically significant. Surprisingly, we found lower scores during the pandemic than in a previous study conducted at our center [34]. Additionally, a study by Huang et al. reported that patients with EQUAL scores above 10 had a significantly higher survival rate compared to those with scores less than 10 [27]. Our findings demonstrated mean scores of 9.92 and 10.36 in survivors in the overall population and CVC carriers, respectively. These results suggest a lower adherence to management guidelines during the pandemic, which may be related to the high mortality rate observed in the overall pandemic population compared to other studies from the pre-pandemic period [35,36].

In contrast to our previous study [34], echocardiography and performing susceptibility testing were not found to be associated with lower mortality. However, ophthalmic exams, when performed, improved survival rates in our population. Although a recent systematic review suggested that universal ophthalmological examination does not improve outcomes [37], the Infectious Diseases Society of America (IDSA) and the European Confederation of Medical Mycology (ECMM) guidelines both recommend that all patients with candidemia should undergo routine dilated funduscopic exam [19,38]. These recommendations were supported by a recent meta-analysis by Phongkhun et al., which found that ocular manifestations in patients with candidemia were more common than previously reported by the American Academy of Ophthalmology (AAO), which suggested an incidence of less than 0.9% [39]. These discrepancies emphasize that further studies are needed to identify high-risk patients that would benefit the most from this intervention. Furthermore, the removal of CVCs within the first 48h of candidemia was found to be associated with higher survival rates. In fact, all guidelines recommend the removal of CVCs in patients with candidemia as a source control measure [38,40,41]; however, there is no clear evidence supporting early removal of CVCs (prior to 48h post infection) [27,34,42].

In concordance with the IDSA recommendations [19] and previous studies [43], treatment with at least 14 days after the first negative blood culture was associated with higher survival rate. Another notable result of our study is that the majority of our patients with CAC had died before receiving the appropriate duration of antifungal treatment. These results may be due to a delay in the diagnosis of candidemia, especially in the early phases of COVID-19. However, it is very difficult to determine in those patients whether the cause of death was related to the COVID-19 respiratory complications versus the candidemia. 

The 30-day mortality rate in the overall population reached 76.6%. Many studies from the pre-pandemic era showed a lower mortality rate of ICU patients with candidemia, with a mortality of 47% in Spain [35], 54% in Japan [44], and 60.8% in France [36]. Higher mortality rates found during the pandemic could be due to many factors including the emergence of the highly resistant *C. auris* and the near collapse of healthcare systems. In addition, Lebanon experienced an economic collapse in 2019 that coincided with the COVID-19 pandemic and resulted in the exodus of health care workers. This added a significant strain on hospitals, including our facility, where a large number of employees were laid off. Junior staff, including physicians and nurses, were recruited, which could have contributed to the delay in the identification and the proper management of patients with candidemia and the lack of adherence to proper infection control practices. Many studies showed a high mortality rate in patients with *C. auris* candidemia [45], reaching 75% in a recent study from our center [8]. Additionally, as discussed before, the adherence to guidelines in the management of candidemia was lower during the pandemic, which could have predisposed to the higher mortality rate. Surprisingly, no statistical difference was found when comparing mortality between CAC and non-CAC, in contrast to other published studies [10,46]. It is possible that with an overall mortality rate of 76.6%, it is difficult to distinguish candidemia attributable mortality from death caused by underlying diseases. Previous studies have suggested that the mortality attributed to candidemia is not significant in a population of patients with high expected mortality [47]. 

In this study, it was found that 80% of patients with CAC were admitted to the ICU, with a median time from ICU admission to candidemia of 17.5 days. This is consistent with previous reports demonstrating that prolonged ICU stay, regardless of COVID-19 status, can increase the risk of developing candidemia [48], and it confirms that the majority of our CAC cases are healthcare-associated. Critically ill COVID-19 patients have been shown to require an average of 20.6 days of ICU admission, further increasing their predisposition to invasive candidiasis [49]. On the other hand, the median time from COVID-19 infection to the development of candidemia in our population was 24 days. Candidemia seems to occur during the second phase of COVID-19 infection, which typically starts after one week of the onset of illness and is characterized by excessive inflammation [50], during which the patients often require steroids and immunomodulatory therapies such as tocilizumab [51], which further increase the risk of invasive fungal infections [48]. These findings highlight the importance of close monitoring and early detection of fungal infections in COVID-19 patients, particularly during the second phase of the COVID-19 infection.

Our study has some limitations. First, it is a retrospective single-center study that may not accurately represent the national epidemiology of candidemia. Second, because all included patients had candidemia, we lacked a comparator group of patients with COVID-19 without candidemia, limiting our assessment of COVID-19 as an independent risk factor for candidemia. We also could not clearly investigate the association between different COVID-19 treatment modalities, especially tocilizumab and CAC, due to the small sample size. In addition, the management of COVID-19 was constantly evolving during the two years of the pandemic, which has led to different treatment protocols over time and may have contributed to improved outcomes. In addition, our study was unable to include MICs data due to financial constraints arising from the economic crisis in Lebanon and the impact of the COVID-19 pandemic. Performing MIC testing on several patients proved to be challenging under these circumstances. Another limitation is that survivorship bias may have influenced the treatment duration, potentially impacting the interpretation of the results. Finally, we acknowledge the limitations of the Vitek2 AST-YS08 method, as specified by the manufacturer; specifically, there are limitations when it comes to certain antibiotic/organism combinations, such as Caspofungin for *C. glabrata* and Fluconazole for *C. glabrata*, *C. kefyr*, and *Cryptococcus neoformans*. These limitations should be taken into consideration when interpreting the results obtained through this microbiological detection method.

## 5. Conclusions

Candidemia is associated with high morbidity and mortality even when treated adequately. The epidemiology of *Candida* spp. in a tertiary care center in Lebanon has significantly shifted during the COVID-19 pandemic, with the emergence of highly resistant spp. including *C. auris*. This shift is alarming as it significantly impacts the management and outcomes of patients. Rigorous infection control practices in combination with antifungal stewardship are essential to containing institutional outbreaks of highly resistant *Candida* spp., but also to decreasing the emergence of future resistance. Institutional epidemiological data are vital to optimizing the choices of agents to be used for empirical antifungal therapy.

## Figures and Tables

**Figure 1 jof-09-00769-f001:**
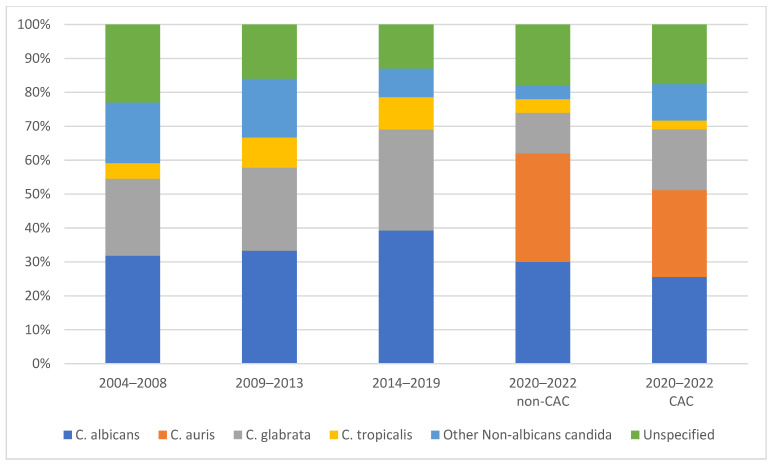
Predominance of *Candida* spp. over 18 years at AUBMC.

**Table 1 jof-09-00769-t001:** Evolution of *Candida* spp. susceptibility in the number of tested isolates over 18 years at AUBMC.

	2004–2008	2009–2013	2014–2019	2020–2022 (Non-CAC)	2020–2022 (CAC)	Total
*C. albicans*						
Fluconazole	1/1 (100%)	6/6 (100%)	13/16 (81.2%)	5/5 (100%)	8/8 (100%)	33/36 (91.6%)
Voriconazole	-	3/4 (75%)	14/17 (82.3%)	5/5 (100%)	8/8 (100%)	30/34 (88.2%)
Amphotericin B	-	3/3 (100%)	17/17 (100%)	5/5 (100%)	8/8 (100%)	33/33 (100%)
Caspofungin	-	1/1 (100%)	6/6 (100%)	5/5 (100%)	8/8 (100%)	20/20 (100%)
*C. tropicalis*						
Fluconazole	2/2 (100%)	-	3/3 (100%)	-	1/1 (100%)	6/6 (100%)
Voriconazole	-	4/5 ((80.0%)	7/7 (100%)	1/1 (100%)	2/2 (100%)	14/15 (93.3%)
Amphotericin B	-	1/1 (100%)	7/7 (100%)	1/1 (100%)	2/2 (100%)	11/11 (100%)
Caspofungin	-	1/1 (100%)	1/1 (100%)	1/1 (100%)	2/2 (100%)	5/5 (100%)
*C. glabrata*						
Fluconazole	3/4 (75%)	3/10 (30.0%)	17/23 (73.9%)	2/6 (33.3%)	4/7 (57.1%)	29/50 (58.0%)
Voriconazole	0/1 (0.0%)	7/8 (87.5%)	18/23 (78.2%)	6/6 (100%)	7/7 (100%)	38/45 (84.4%)
Amphotericin B	-	1/1 (100%)	22/23 (95.6%)	6/6 (100%)	6/6 (100%)	35/36 (97.2%)
Caspofungin	-	-	9/9 (100%)	3/6 (50.0%)	2/7 28.5%)	14/22 (63.6%)
*C. parapsilosis*						
Fluconazole	2/2 (100%)	-	3/3 (100%)	-	1/1 (100%)	6/6 (100%)
Voriconazole	-	-	3/3 (100%)	-	1/1 (100%)	4/4 (100%)
Amphotericin B	-	-	3/3 (100%)	-	1/1 (100%)	4/4 (100%)
Caspofungin	-	-	-	-	1/1 (100%)	1/1 (100%)
*C. auris*						
Fluconazole	-	-	-	0/3 (0.0%)	2/9 (22.2%)	2/12 (16.7%)
Voriconazole	-	-	-	1/3 (33.3%)	5/9 (55.5%)	6/12 (50.0%)
Amphotericin B	-	-	-	0/3 (0.0%)	0/7 (0.0%)	0/10 (0.0%)
Caspofungin	-	-	-	8/8 (100%)	3/3 (100%)	11/11 (100%)

**Table 2 jof-09-00769-t002:** Characteristics of episodes of CAC and non-CAC in critically ill patients. ESRD, end-stage renal disease; HD, hemodialysis; AKI, acute kidney injury; CVC, central venous catheter; CLABSI, central line associated bloodstream infection; UTI, urinary tract infection; GI, gastrointestinal.

	Total N = 64	CAC N = 32 (50.0%)	Non-CAC N = 32 (50.0%)	*p*-Value
Age *	73 (19)	75 (18)	72 (18)	0.14
Male	38 (59.4%)	20 (62.5%)	18 (56.3%)	0.61
Diabetes mellitus	29 (45.3%)	18 (56.3%)	11 (34.4%)	0.07
ESRD on HD	26 (40.6%)	12 (37.5%)	14 (43.8%)	0.61
AKI requiring HD	2 (3.1%)	1 (3.1%)	1 (3.1%)	1.00
Hematologic malignancy	8 (12.5%)	5 (15.6%)	3 (9.4%)	0.70
Solid organ malignancy	16 (25.0%)	5 (15.6%)	11 (34.4%)	0.08
Recent chemotherapy	14 (22.2%)	4 (12.5%)	10 (32.3%)	0.05
Recent immunotherapy	4 (6.3%)	2 (6.3%)	2 (6.3%)	1.00
Neutropenia	4 (6.3%)	1 (3.1%)	3 (9.4%)	0.61
Recent abdominal surgery **	6 (9.4%)	2 (6.3%)	4 (12.5%)	0.67
Recent antibiotics **	62 (96.9%)	30 (93.8%)	32 (100.0%)	0.49
Recent antifungals **	22 (34.4%)	12 (37.5%)	10 (31.3%)	0.59
Mechanical ventilation	49 (76.6%)	26 (81.3%)	23 (71.9%)	0.37
CVC	55 (85.9%)	27 (84.4%)	28 (87.5%)	1.00
Persistent candidemia	10 (24.4%)	3 (15.8%)	7 (31.8%)	0.29
Source of candidemia				
CLABSI	18 (28.1%)	8 (25.0%)	10 (31.2%)	0.57
UTI	20 (31.3%)	12 (37.5%)	8 (25.0%)	0.28
GI tract	14 (21.9%)	6 (18.8%)	8 (25.0%)	0.54
Unknown	15 (23.4%)	9 (28.1%)	6 (18.8%)	0.37
Others ***	11 (17.5%)	4 (12.9%)	7 (21.9%)	0.34
Species				0.94
*C. albicans*	17 (26.6%)	9 (28.1%)	8 (25.1%)	
*C. auris*	19 (29.7%)	9 (28.1%)	10 (31.3%)	
NAC other than *C. auris*	28 (43.8%)	14 (43.8%)	14 (43.8%)	

* Median (IQR).** 30 days prior to candidemia *** Vertebral abscess, aortoiliac graft infection, thigh necrotic ulcer.

**Table 3 jof-09-00769-t003:** Management and outcomes of CAC and non-CAC episodes in critically-ill patients.

	Total N = 64	CAC N = 32 (50.0%)	Non-CAC N = 32 (50.0%)	*p*-Value
Speciation	51 (79.7%)	26 (81.3%)	25 (78.1%)	0.75
Susceptibility testing	28 (48.3%)	15 (51.7%)	13 (44.8%)	0.59
Echocardiography	15 (24.2%)	9 (29.0%)	6 (19.4%)	0.37
Ophthalmic examination	11 (17.5%)	5 (15.6%)	6 (19.4%)	0.69
Empiric antifungal agent				
Fluconazole	3 (5.1%)	2 (6.7%)	1 (3.4%)	1.00
Caspofungin	36 (61.0%)	25 (83.3%)	11 (37.9%)	<0.001
Anidulafungin	14 (23.7%)	5 (16.7%)	9 (31.0%)	0.19
Micafungin	1 (1.7%)	0 (0.0%)	1 (3.4%)	0.49
Lipid formulation of Amphotericin B	5 (8.5%)	1 (3.3%)	4 (13.8%)	0.19
Voriconazole	7 (11.9%)	3 (10.0%)	4 (13.8%)	0.71
Targeted antifungal agent				0.054
Fluconazole	44 (47.3%)	3 (23.1%)	41 (51.2%)	
Voriconazole	14 (15.1%)	3 (23.1%)	11 (13.8%)	
Caspofungin	13 (14%)	2 (15.4%)	11 (13.8%)	
Anidulafungin	10 (10.8%)	4 (30.8%)	6 (7.5%)	
Micafungin	4 (4.3%)	1 (7.7%)	3 (3.8%)	
Lipid formulation of Amphotericin B	8 (8.6%)	0 (0.0%)	8 (10.0%)	
Empirical echinocandin	48 (81.4%)	28 (93.3%)	20 (69.0%)	0.016
Step-down to fluconazole	4 (7.1%)	1 (3.3%)	3 (11.5%)	0.32
Daily blood culture until negative	2 (4.2%)	1 (4.2%)	1 (4.2%)	1.00
Completed 14 days of antifungals	9 (17.0%)	4 (14.8%)	5 (19.2%)	0.72
30-day mortality	49 (76.6%)	24 (75.0%)	25 (78.1%)	0.76

## Data Availability

The data that support the findings of this study are available on request from the corresponding author. The data are not publicly available due to privacy or ethical restrictions.

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
