# Peer review of "The Impact of COVID-19 on the Epidemiology and Outcomes of Candidemia: A Retrospective Study from a Tertiary Care Center in Lebanon"

_jof, 2023, doi:10.3390/jof9070769_

Round 1
Reviewer 1 Report
This is a well-written, interesting article that compares the epidemiology of Candida bloodstream infections in patients with and without COVID-19. Overall, the manuscript is clearly presented and easy to understand. I have several suggestions that may improve the clarity of the manuscript.
Major comments
1. Line 42: The study of Paiva et al. published in 2016 refers to characteristics of the hospital-acquired fungemias in ICUs, not in the general population. Of note, according to the results of the recent ECMM Candida III Multinational European study C. albicans is not the predominant spp. as it accounts for 44% (276/632) of candidemias in Europe (Hoenigl et al. Open Forum Infectious Diseases, Volume 9, Issue Supplement_2, December 2022, ofac492.038, https://doi.org/10.1093/ofid/ofac492.038). Please rephrase and correct.
2. Lines 81-82 and 122. The CLSI does not provide clinical breakpoints for all Candida spp. and antifungals, for example there are not available clinical breakpoints for amphotericin B. In such cases, did you use epidemiological cut-off values to discriminate between wild type and non-wild type isolates? Furthermore, tentative breakpoints for C. auris are currently provided for fluconazole, echinocandins and amphotericin B by the CDC (https://www.cdc.gov/fungal/candida-auris/c-auris-antifungal.html), while the CLSI has recently published epidemiological cut-off values only for echinocandins (M57S). The criteria used for the interpretation of the results should be clearly stated. Please rephrase and correct.
3. Line 120: Please indicate the version of Vitek2 used for the identification of isolates. Of note, the Vitek2 version 8.01 was found to consistently identify some C. auris, but not all clades, thus accurate identification of C. auris is currently proposed either by MALDI-TOF MS or molecular methods (https://www.cdc.gov/fungal/candida-auris/identification.html). This should be further commented in the discussion section as a potential limitation of the study.
4. Line 120: Please indicate the version of Vitek2 card used for antifungal susceptibility testing. Vitek2 is not currently proposed for antifungal susceptibility testing of C. auris. For example, the CLSI has recently suggested that amphotericin B susceptibility results should be interpreted with caution because of the significant variability in MIC values across different testing methodologies with Vitek2 being more likely to report isolates having amphotericin B MICs >2 mg/L, i.e. false resistance (https://clsi.org/about/blog/ast-news-update-june-2022-hot-topic/). The aforementioned in conjunction with the fact that you did not verify the resistant phenotypes with a reference method should be further commented in the discussion section as an important limitation of the study.
5. Lines 125: How was the 7-day time frame for recurrent candidemia selected? Please provide reference. Based on the NHSN definitions (line 126), the repeat infection timeframe is a 14-day time frame during which no new infections of the same type are reported, while the majority of epidemiological surveys use a 30-day time frame to define recurrent cases. Could the short-time frame used in the present study have overestimated the incidence of episodes?
6. Lines 147-149: What about differences in the annual incidence rate of candidemia throughout the study period? It would be of interest to compare the number of episodes/year and see whether the pandemic affected significantly their occurrence.
7. Line 163: Was the decrease in C. glabrata rate statistically significant?
8. Lines 163-166: Could the emergence of C. auris candidemias be attributed to a short-time outbreak? Were the C. auris isolates equally distributed during the 2-year period of the pandemic? Intra-hospital C. auris transmission cannot be ruled out since genotyping of the recovered isolates was not performed. This should be further commented in the results and particularly in discussion section.
9. Lines 172-178: Based on comment 2, some of the isolates should have been classified as wild type or non-wild type instead of susceptible and resistant, respectively, in terms of their antifungal susceptibility. Please rephrase and correct throughout.
10. Table 1: Please present the rate of resistant/non-wild-type isolates, i.e. percentages instead of number, converting its content more reader-friendly. Please also correct its footnote regarding “susceptibility” based on the aforementioned comment.
11. Lines 177-178 and Table 1: How was the rate of voriconazole-resistant C. auris isolates calculated? Of note, neither the CDC (https://www.cdc.gov/fungal/candida-auris/c-auris-antifungal.html) nor the CLSI (M57S) provides tentative and epidemiological cut-off values, respectively. Please rephrase and correct.
12. Lines 258-261: Please note that the manufacturer indicates the limitation of Vitek2 AST-YS08 and suggests performing an alternative method of testing prior to reporting of results for the following antibiotic/organism combination(s):
• Caspofungin: Candida glabrata (when applying CLSI breakpoints <=0.12 S, 0.25 I, >=0.5 R)
• Fluconazole: Candida glabrata, C. kefyr, Cryptococcus neoformans
This should be further commented in the discussion section.
13. Line 258: Based on Table 1, the rate of fluconazole-resistant C. grabrata strains before the pandemic was 62% (23/37 isolates). Please correct and indicate whether this difference was statistically significant.
14. Lines 29-30, 231-233 and 311: According to the present findings, the 30-day mortality rate of critically ill (ICU) patients with/without CAC between March 2020 and March 2022 was 75-78%. Nevertheless, in your previous study you included all patients not only ICU in the 30-day crude mortality rate analysis reporting 55% (https://pubmed.ncbi.nlm.nih.gov/33477771/). Since the data are not comparable, please rephrase and correct throughout the manuscript.
15. Line 229: Please consider comments 4 and 11 regarding the “higher resistance rates” and correct accordingly.
Minor comments
1. Line 46: Please capitalize the word Candida.
2. Line 67: Please rephrase “the outcomes of patients with CAC and those with candidemia but not infected with COVID-19” instead of “the outcomes of patients with CAC and those with candidemia in patients not infected with COVID-19”.
3. Line 138: P instead of α. Please correct.
4. When an abbreviation/acronym is defined in the manuscript upon its first use, it should be used throughout. For example lines 116 and 117 since NAC is already defined in line 46, line 151 since ICU is already defined in line 62, lines 243, 247, 263 since MDR is already defined in line 50 etc. Please correct throughout.
5. Similarly, please consider using either species or spp. throughout the manuscript.
Author Response
Dear Editor,
We would like to express our gratitude for allowing us to resubmit a revised version of our manuscript. We thank the reviewers for meticulously reviewing and providing their extremely valuable comments. We believe those comments have significantly improved the quality of our manuscript. We hereby provide a point-by-point answer to the reviewer’s comments.
Major comments
- Line 42: The study of Paiva et al. published in 2016 refers to characteristics of the hospital-acquired fungemias in ICUs, not in the general population. Of note, according to the results of the recent ECMM Candida III Multinational European study C. albicans is not the predominant spp. as it accounts for 44% (276/632) of candidemias in Europe (Hoenigl et al. Open Forum Infectious Diseases, Volume 9, Issue Supplement_2, December 2022, ofac492.038, https://doi.org/10.1093/ofid/ofac492.038). Please rephrase and correct.
We thank the reviewer for their valuable comment. Indeed, it is essential that comparisons be made between comparable populations. Although most studies do mention that Candida albicans is the predominant species when compared to other species individually, it is not predominant when compared to non-albicans species combined. We have modified the sentence in line 48-49 to the following and have added the provided reference by Hoenigl et al. “Studies from Lebanon show a predominance of non-albicans Candida (NAC) spp. in episodes of candidemia [4], which is similar to what has been reported from European countries [5] and other neighboring countries from the Middle East region [3, 6, 7]”
- Lines 81-82 and 122. The CLSI does not provide clinical breakpoints for all Candida spp. and antifungals, for example there are not available clinical breakpoints for amphotericin B. In such cases, did you use epidemiological cut-off values to discriminate between wild type and non-wild type isolates? Furthermore, tentative breakpoints for C. auris are currently provided for fluconazole, echinocandins and amphotericin B by the CDC (https://www.cdc.gov/fungal/candida-auris/c-auris-antifungal.html), while the CLSI has recently published epidemiological cut-off values only for echinocandins (M57S). The criteria used for the interpretation of the results should be clearly stated. Please rephrase and correct.
We thank the reviewer for their important feedback. Indeed, clinical breakpoints for amphotericin B are yet to be established by the CLSI. The microbiology laboratory at AUBMC uses both the CLSI and CDC breakpoints to interpret strains’ susceptibilities defined as follows: fluconazole, ≥32 μg/mL; caspofungin, ≥2 μg/mL; micafungin, ≥4 μg/mL; amphotericin B, ≥2 μg/mL. We have added the following statement to lines 133-135 in the “Microbiological definitions” subsection: “The microbiology laboratory used the Clinical and Laboratory Standards Institute breakpoints (CLSI) [16], and the Centers for Disease Control and Prevention (CDC) tentative MIC cut-off values that are based on data from similar Candida spp. {CDC, 2020 #408} to interpret strains’ susceptibilities against amphotericin B, caspofungin, micafungin, fluconazole, and voriconazole. The resistance breakpoints were defined as follows: fluconazole, ≥32 μg/mL; caspofungin, ≥2 μg/mL; micafungin, ≥4 μg/mL; amphotericin B, ≥2 μg/mL. Resistant breakpoints to voriconazole were extrapolated from fluconazole. Due to the financial issues in view of the economic crisis in Lebanon and the COVID-19 pandemic, minimal inhibitory concentrations (MICs) were not performed for all patients because of their high cost.”
- Line 120: Please indicate the version of Vitek2 used for the identification of isolates. Of note, the Vitek2 version 8.01 was found to consistently identify some C. auris, but not all clades, thus accurate identification of C. auris is currently proposed either by MALDI-TOF MS or molecular methods (https://www.cdc.gov/fungal/candida-auris/identification.html). This should be further commented in the discussion section as a potential limitation of the study.
We thank the reviewer for their important comment. Both Matrix-Assisted Laser Desorption/Ionization-Time of Flight MALDI-TOF (Bruker Daltonik, GmbH, Bremen, Germany) and Vitek2 were used for the identification of C. auris species. We have added the following statement in line 130 in the methods section: “Additionally, C. auris colonies were phenotypically identified by Matrix-Assisted Laser Desorption/Ionization-Time of Flight MALDI-TOF (Bruker Daltonik, GmbH, Bremen, Germany).”
- Line 120: Please indicate the version of Vitek2 card used for antifungal susceptibility testing. Vitek2 is not currently proposed for antifungal susceptibility testing of C. auris. For example, the CLSI has recently suggested that amphotericin B susceptibility results should be interpreted with caution because of the significant variability in MIC values across different testing methodologies with Vitek2 being more likely to report isolates having amphotericin B MICs >2 mg/L, i.e. false resistance (https://clsi.org/about/blog/ast-news-update-june-2022-hot-topic/). The aforementioned in conjunction with the fact that you did not verify the resistant phenotypes with a reference method should be further commented in the discussion section as an important limitation of the study.
We thank the reviewer for their valuable input. Antifungal susceptibility testing was done using VITEK-2 and E-test antimicrobial susceptibility tests. We added the following statement in line 133-134: “Antifungal susceptibility testing was done using VITEK-2 and E-test antimicrobial susceptibility tests.”
- Lines 125: How was the 7-day time frame for recurrent candidemia selected? Please provide reference. Based on the NHSN definitions (line 126), the repeat infection timeframe is a 14-day time frame during which no new infections of the same type are reported, while the majority of epidemiological surveys use a 30-day time frame to define recurrent cases. Could the short-time frame used in the present study have overestimated the incidence of episodes?
We thank the reviewer for their valuable comment. Although we mentioned that recurrent candidemia was defined as occurring after 7 days, none of the patients who had candidemia after 2020 and were included in the survival analysis had recurrent candidemia. We have corrected the definition of recurrent candidemia according to the NHSN definitions in line 153 and have rechecked our data to identify if any patients had recurrent candidemia according to the newer definition. We found that none of the patients had recurrence during the first 14 days. We have mentioned that none of the patients had recurrent candidemia in line 215:” None of the patients were found to have recurrent candidemia.”
- Lines 147-149: What about differences in the annual incidence rate of candidemia throughout the study period? It would be of interest to compare the number of episodes/year and see whether the pandemic affected significantly their occurrence.
We extend our appreciation to the reviewer for their insightful comment. Unfortunately, we were unable to determine the annual incidence rates of candidemia in adult patients as we were not able to acquire the annual patient days for adult patients. Therefore, the annual incidence of candidemia per year cannot be determined due to data limitations.
- Line 163: Was the decrease in C. glabrata rate statistically significant?
We would like to express our gratitude to the reviewer for their valuable comment. We reported the prevalence of Candida glabrata and not its incidence rate as we were unable to determine the annual incidence given that annual patient days numbers were not obtainable. In light of the small number of episodes, we did not perform statistical tests as statistical significance (if determined) would not be clinically relevant.
- Lines 163-166: Could the emergence of C. auris candidemias be attributed to a short-time outbreak? Were the C. auris isolates equally distributed during the 2-year period of the pandemic? Intra-hospital C. auris transmission cannot be ruled out since genotyping of the recovered isolates was not performed. This should be further commented in the results and particularly in discussion section.
We thank the reviewer for their comment. The highest rate of C. auris candidemia was observed in 2021, with one episode occurring in 2020, 23 episodes in 2021, and two episodes between January and March 2022. The 23 episodes were distributed across the entire year 2021, indicating an outbreak that was effectively managed by the infection control team. We added the following statement in the discussion section to emphasize on this point in line 279:” At our center, a significant number of C. auris candidemia cases were observed in 2021, indicating the occurrence of an outbreak resulting from intra-hospital transmission. Genome sequencing analysis to determine clade distribution was performed on 29 isolates from different culture sites (blood, central venous catheter, deep tracheal aspiration, urine…) {Reslan, 2022 #412} and confirmed that all C. auris genomes belonged to South Asian clade I among the five known global clades regardless of the recovered source, site of specimen, or time span between isolations, thereby highly reflecting an outbreak due to hospital-associated transmission. However, due to financial constraints caused by the economic crisis in Lebanon and the ongoing COVID-19 pandemic, conducting genome sequencing analysis on all isolates was not possible.”
- Lines 172-178: Based on comment 2, some of the isolates should have been classified as wild type or non-wild type instead of susceptible and resistant, respectively, in terms of their antifungal susceptibility. Please rephrase and correct throughout.
We thank the reviewer for their input. However, since we added the CDC cutoffs used in our microbiology lab to the methods which use the terms “susceptible/resistant” rather than “wild type/non-wild type”, we have elected to maintain the use of resistance/resistant since no epidemiological cut-off value were used in our study. Additionally, MICs were not collected in our data due to the due to the lack of resources during the economic crisis that occurred in Lebanon during pandemic. This clarification will enhance the clarity and consistency of our results, as no epidemiological cutoff values were utilized in our study. We added a statement to line 402 to develop that as a limitation to our study “In addition, we were unable to report MICs due to financial constraints arising from the economic crisis in Lebanon and the impact of the COVID-19 pandemic. Conducting minimal inhibitory concentration (MIC) testing on several patients proved to be challenging and costly under these circumstances.”
- Table 1: Please present the rate of resistant/non-wild-type isolates, i.e. percentages instead of number, converting its content more reader-friendly. Please also correct its footnote regarding “susceptibility” based on the aforementioned comment.
We thank the reviewer for their valuable comment. Although percentages are much more reader friendly than frequencies, we have initially elected to present the data in frequencies to highlight the number of isolates that were tested as one isolate of C. albicans from 2004-2008 may not be representative of the entire isolates. We have added the respective percentages in between brackets.
- Lines 177-178 and Table 1: How was the rate of voriconazole-resistant C. auris isolates calculated? Of note, neither the CDC (https://www.cdc.gov/fungal/candida-auris/c-auris-antifungal.html) nor the CLSI (M57S) provides tentative and epidemiological cut-off values, respectively. Please rephrase and correct.
We thank the reviewer for their valuable input. As previously mentioned, the resistance breakpoints to voriconazole were extrapolated from fluconazole We have added the following statements to the microbiological methods in line 141 to clarify: “Resistant breakpoints to voriconazole were extrapolated from fluconazole.” Moreover, due to the high cost and non-routine nature of obtaining Minimum Inhibitory Concentrations (MICs), we are unable to report the data with complete accuracy and precision.
- Lines 258-261: Please note that the manufacturer indicates the limitation of Vitek2 AST-YS08 and suggests performing an alternative method of testing prior to reporting of results for the following antibiotic/organism combination(s):
- Caspofungin: Candida glabrata (when applying CLSI breakpoints <=0.12 S, 0.25 I, >=0.5 R)
- Fluconazole: Candida glabrata, C. kefyr, Cryptococcus neoformans
This should be further commented in the discussion section.
We thank the reviewer for bringing this to our attention. We acknowledge the limitations of the Vitek2 AST-YS08 method as indicated by the manufacturer, particularly regarding specific antibiotic/organism combinations such as Caspofungin for Candida glabrata and Fluconazole for Candida glabrata, C. kefyr, and Cryptococcus neoformans. This was added to the limitations section to line 402 as follows: “Finally, we acknowledge the limitations of the Vitek2 AST-YS08 method, as specified by the manufacturer. Specifically, there are limitations when it comes to certain antibiotic/organism combinations, such as Caspofungin for C. glabrata and Fluconazole for C.glabrata, C. kefyr, and Cryptococcus neoformans. These limitations should be taken into consideration when interpreting the results obtained through this microbiological detection method.”
- Line 258: Based on Table 1, the rate of fluconazole-resistant C. grabrata strains before the pandemic was 62% (23/37 isolates). Please correct and indicate whether this difference was statistically significant.
We appreciate the reviewer’s comment. This percentage was corrected in line 306: “Susceptibility rates to fluconazole in C. glabrata isolates decreased from 62% before the pandemic to 46.1% during the pandemic”.
The reported susceptibility rate was presented as the number of episodes per year, without conducting a statistical comparison between different years due to the relatively small number of episodes.
- Lines 29-30, 231-233 and 311: According to the present findings, the 30-day mortality rate of critically ill (ICU) patients with/without CAC between March 2020 and March 2022 was 75-78%. Nevertheless, in your previous study you included all patients not only ICU in the 30-day crude mortality rate analysis reporting 55% (https://pubmed.ncbi.nlm.nih.gov/33477771/). Since the data are not comparable, please rephrase and correct throughout the manuscript.
We thank the reviewer for their valuable comment. In comparison to other studies from the literature who reported patients with CAC in ICU we added several studies, and we removed the study conducted at our center since it included all patients, and corrected throughout:
- Line 329: “which may be related to the high mortality rate observed in the overall pandemic population compared to other studies from the pre-pandemic period [34, 35].”
- Lines 359: We included the mortality rates from studies including candidemia in ICU populations: “The 30-day mortality rate in the overall population reached 76.6%. Many studies from the pre-pandemic era showed a lower mortality rate of ICU patients with candidemia with a mortality of 47% in Spain [34], 54% in Japan [43], and 60.8% in France [35].”
- Lines 30-31: Adjusted the sentence as follows: “Mortality rate in the overall ICU population during the pandemic was 76.6%, much higher than the previously reported candidemia mortality rate observed in studies involving ICU patients.” instead of “much higher than the previously reported mortality of candidemia from previous studies at our center”
- Lines 264: Adjusted the sentence as follows “higher mortality rate of patients with candidemia during the pandemic compared to studies from the pre-pandemic” instead of “higher mortality rate of patients with candidemia during the pandemic compared to the pre-pandemic period”
- Line 229: Please consider comments 4 and 11 regarding the “higher resistance rates” and correct accordingly.
We thank the reviewer for their important input. Based on our responses to comments 4 and 11, as well as the information provided in the methods section where we utilized VITEK-2 and E-test antimicrobial susceptibility methods, we believe it would be appropriate to use the term "higher resistance rate" in our manuscript.
Minor comments
- Line 46: Please capitalize the word Candida.
We thank the reviewer for their valuable comment. This was corrected in line 47.
- Line 67: Please rephrase “the outcomes of patients with CAC and those with candidemia butnot infected with COVID-19” instead of “the outcomes of patients with CAC and those with candidemia in patients not infected with COVID-19”.
We thank the reviewer for their valuable comment. This was corrected in line 70-71.
- Line 138: P instead of α. Please correct.
We thank the reviewer for their valuable comment. This was corrected in line 167.
- When an abbreviation/acronym is defined in the manuscript upon its first use, it should be used throughout. For example lines 116 and 117 since NAC is already defined in line 46, line 151 since ICU is already defined in line 62, lines 243, 247, 263 since MDR is already defined in line 50 etc. Please correct throughout.
We thank the reviewer for their valuable comment. This was corrected throughout the manuscript.
- Similarly, please consider using either species or spp. throughout the manuscript.
We thank the reviewer for their valuable comment. This was corrected throughout the manuscript.
We thank the reviewers again for providing their valuable comments which have significantly improved the quality of this manuscript. We look forward to your decision.
Sincerely,
Souha Kanj, MD, FACP, FIDSA, FRCP, FESCMID, FECMM
Professor of Medicine,
Associate Vice President for Global Affairs,
Head, Division of Infectious Diseases,
Chairperson, Infection Control Program
Co-director, Antimicrobial Stewardship Program
American University of Beirut Medical Center
Consulting Professor, Duke University Medical Center, NC, USA,
Honorary Dr., Radboud University, Netherlands.

Reviewer 2 Report
The MS entitle "the impact of COVID 19 on the epidmemiology and outcome of Candidemia by Zakhem et al.provides interesting data ;however some revisions require to clarify and improve the quality of this research.
Introduction:please cite some other related articles that reported NAC for Candidemia in middle East countries.
Methods:
The identification of Candida species just performed VITEK in the blood culture specimens ?How about molecular identification ?How did you confirm C.auris in your study ?If yes with which method ?
Did the authors perform AFST according CLSI M27/s4 ?In this case , please add the relevant reference there.
Result: in Table 1 authors must add MIC50 ,MIC 90 and GM for each antifungals .
Discussion :
Please add this reference :Candidemia among Iranian patients with COVID-19 addmitted to ICU by Amir Arastehfar as a comparision of distribution and AFST findings among Candida isolates of candidemia in COVID-19 patients .
The MS is written well in terms of English Language .
Author Response
Dear Editor,
We would like to express our gratitude for allowing us to resubmit a revised version of our manuscript. We thank the reviewers for meticulously reviewing and providing their extremely valuable comments. We believe those comments have significantly improved the quality of our manuscript. We hereby provide a point-by-point answer to the reviewer’s comments.
The MS entitle "the impact of COVID 19 on the epidmemiology and outcome of Candidemia by Zakhem et al.provides interesting data ;however some revisions require to clarify and improve the quality of this research.
Introduction:please cite some other related articles that reported NAC for Candidemia in middle East countries.
We thank the the reviewer for their valuable comment. More articles from the MENA region were added in lines 48-49 as follows: “which is similar to what has been reported from European countries [5] and other neighboring countries from the MENA region [3, 6, 7]”
Methods:
The identification of Candida species just performed VITEK in the blood culture specimens ?How about molecular identification ?How did you confirm C.auris in your study ?If yes with which method ?
We thank the reviewer for their important comment. Both Matrix-Assisted Laser Desorption/Ionization-Time of Flight MALDI-TOF (Bruker Daltonik, GmbH, Bremen, Germany) and Vitek2 were used for the identification of C. auris species. We have added the following statement in line 130 in the methods section: “Additionally, C. auris colonies were phenotypically identified by Matrix-Assisted Laser Desorption/Ionization-Time of Flight MALDI-TOF (Bruker Daltonik, GmbH, Bremen, Germany).”
Did the authors perform AFST according CLSI M27/s4 ? In this case , please add the relevant reference there.
We appreciate the reviewer’s feedback. This reference was added in line 134: “Antifungal susceptibility testing was done using VITEK-2 and E-test antimicrobial susceptibility tests. The microbiology laboratory used the Clinical and Laboratory Standards Institute breakpoints (CLSI) [17, 18], and the Centers for Disease Control and Prevention (CDC) tentative MIC cut-off values that are based on data from similar Candida spp. {CDC, 2020 #408} to interpret strains’ susceptibilities against amphotericin B, caspofungin, micafungin, fluconazole, and voriconazole”
Result: in Table 1 authors must add MIC50 ,MIC 90 and GM for each antifungals .
We thank the reviewer for their valuable input. Unfortunately, due to the high cost and non-routine nature of obtaining Minimum Inhibitory Concentrations (MICs), we are unable to report the data with complete accuracy and precision. This was further discussed in the limitations in the discussion section in line 402 as follows: “In addition, our study was unable to include MICs due to financial constraints arising from the economic crisis in Lebanon and the impact of the COVID-19 pandemic. Conducting minimal inhibitory concentration (MIC) testing on several patients proved to be challenging under these circumstances.”
Discussion :
Please add this reference :Candidemia among Iranian patients with COVID-19 addmitted to ICU by Amir Arastehfar as a comparision of distribution and AFST findings among Candida isolates of candidemia in COVID-19 patients .
We thank the reviewer for their valuable input. The reference was added in the discussion in lines 272 “This observation may differ from other studies conducted in Iran and Oman, where C. albicans and C. glabrata remained the predominant pathogens causing candidemia during the pandemic”.
We thank the reviewers again for providing their valuable comments which have significantly improved the quality of this manuscript. We look forward to your decision.
Sincerely,
Souha Kanj, MD, FACP, FIDSA, FRCP, FESCMID, FECMM
Professor of Medicine,
Associate Vice President for Global Affairs,
Head, Division of Infectious Diseases,
Chairperson, Infection Control Program
Co-director, Antimicrobial Stewardship Program
American University of Beirut Medical Center
Consulting Professor, Duke University Medical Center, NC, USA,
Honorary Dr., Radboud University, Netherlands.

Reviewer 3 Report
It is a study that proposed to " compare the outcomes of paients with CAC and those with candidemia in patients not infected with COVID-19" in a not well defined group of patients". Although it goes for a wide time interval, it does not present with a statistically valid method to look for the differences between the groups proposed.
I think that could be published as a description of the Candida bloodstream episodes in that center but it has a strong limitation to evaluate the differences between those groups and the relationship with Covid.
2. Research design. The type of study is not well defined, is it a cohort, a cross sectional?
The outcome is not adequately defined. There is no formal definition. The aim suggests to evaluate mortality or hospital stay, however in the statistical section the proposed outcome variable is the presence of a CAC. In that sense the study should be the predictors of CAC and the design would be quite different. Since no clear design is presented, there is no clarity on the design and methodology.
On the other hand, the sample size is too small to allow to identify a clear difference between the two groups (CAC, non CAC). It would need differences of more than 20% between the two groups to detect a statistically difference.
Line 94: The following text can not be understand: "We did not include patients without COVID-19 who had candidemia prior to the onset of the pandemic to adjust to the ongoing changes of the fungal ecology at our center." It might be read as any case of candidemia previous the pandemic was excluded.
Line 131: No statistical method is presented to validate a little more than the plain differences between the groups.
The time duration of the treatment migth be affected by survivorship bias. No appropriate statistical method was used to look for mortality and to adjust for the differences.
None.
Author Response
Dear Editor,
We would like to express our gratitude for allowing us to resubmit a revised version of our manuscript. We thank the reviewers for meticulously reviewing and providing their extremely valuable comments. We believe those comments have significantly improved the quality of our manuscript. We hereby provide a point-by-point answer to the reviewer’s comments.
It is a study that proposed to" compare the outcomes of patients with CAC and those with candidemia in patients not infected with COVID-19" in a not well defined group of patients". Although it goes for a wide time interval, it does not present with a statistically valid method to look for the differences between the groups proposed.
I think that could be published as a description of the Candida bloodstream episodes in that center but it has a strong limitation to evaluate the differences between those groups and the relationship with Covid.
- Research design. The type of study is not well defined, is it a cohort, a cross sectional?
The outcome is not adequately defined. There is no formal definition. The aim suggests to evaluate mortality or hospital stay, however in the statistical section the proposed outcome variable is the presence of a CAC. In that sense the study should be the predictors of CAC and the design would be quite different. Since no clear design is presented, there is no clarity on the design and methodology.
On the other hand, the sample size is too small to allow to identify a clear difference between the two groups (CAC, non CAC). It would need differences of more than 20% between the two groups to detect a statistically difference.
We thank the reviewer for meticulously reviewing the manuscript. We believe that one of the main contributions of this study is the lack of available data from Lebanon and the Arab region on COVID-19 associated candidemia (CAC), as well as the impact of the COVID-19 pandemic on the epidemiology of candidemia in this region. This study aims to address this knowledge gap and provide insights into this specific area.
We would like to emphasize that the study design was retrospective, as stated in line 78, and that the main outcome assessed was mortality as stated in line 80 in our methods: “Our main outcome was mortality.
” We acknowledge that hospital stay of patients with CAC was not evaluated, and due to the lack of a comparator group of COVID-19 patients without candidemia, assessing risk factors of CAC could not be done and was beyond the aims of this study which aimed to assess the microbiological profile of candidemia during the pandemic period and identify if CAC was associated with a higher mortality than non-CAC.
Line 131: No statistical method is presented to validate a little more than the plain differences between the groups.
We thank the reviewer for their comment. Considering the small number of patients in the study, it is important to note that statistical methods may not adequately reflect clinical significance, particularly in terms of mortality, especially considering the severity of illness observed in the study population.
Line 94: The following text can not be understand: "We did not include patients without COVID-19 who had candidemia prior to the onset of the pandemic to adjust to the ongoing changes of the fungal ecology at our center." It might be read as any case of candidemia previous the pandemic was excluded.
We thank the reviewer for providing their input. The sentence was edited into the following to avoid confusion in line 102: “To account for the dynamic changes in Candida spp. epidemiology at our center during the pandemic, we focused on comparing outcomes in the subpopulation of patients admitted to ICU only during the pandemic” instead of “We did not include patients without COVID-19 who had candidemia prior to the onset of the pandemic to adjust to the ongoing changes of the fungal ecology at our center.”
The time duration of the treatment might be affected by survivorship bias. No appropriate statistical method was used to look for mortality and to adjust for the differences.
We would like to thank the reviewer for raising this important matter. Indeed, we acknowledge the presence of survivorship bias. We added this to our limitations in line 406 as follows: “Survivorship bias may have influenced the treatment duration, potentially impacting the interpretation of the results.”
We thank the reviewers again for providing their valuable comments which have significantly improved the quality of this manuscript. We look forward to your decision.
Sincerely,
Souha Kanj, MD, FACP, FIDSA, FRCP, FESCMID, FECMM
Professor of Medicine,
Associate Vice President for Global Affairs,
Head, Division of Infectious Diseases,
Chairperson, Infection Control Program
Co-director, Antimicrobial Stewardship Program
American University of Beirut Medical Center
Consulting Professor, Duke University Medical Center, NC, USA,
Honorary Dr., Radboud University, Netherlands.

Round 2
Reviewer 3 Report
Thank you to the authors for carefully reviewing the comments. However I think that in the present state the paper cannot be published. The aim of the study is to compare. I still think that a logistic regression can be done. However if the authors think that it is not possible, no references should be done in the manuscript about comparison. I would suggest to change the aim to just " describe the mortality" instead of "compare the outcomes". I would suggest to write the statistical analysis as "exploratory" since no further statistical consideration was done. I would also eliminate any "Significantly asssociation" with the outcome, since no valid statistical method was done to prove (or discard such asssociation).
